# Discovery of Novel GMPS Inhibitors of *Candidatus* Liberibacter Asiaticus by Structure Based Design and Enzyme Kinetic

**DOI:** 10.3390/biology10070594

**Published:** 2021-06-28

**Authors:** Jing Nan, Shaoran Zhang, Ping Zhan, Ling Jiang

**Affiliations:** 1Ministry of Education Key Laboratory of Plant Biology, Huazhong Agricultural University, Wuhan 430070, China; nanjing@webmail.hzau.edu.cn (J.N.); zhanping818@163.com (P.Z.); 2State Key Laboratory of Agricultural Microbiology, Huazhong Agricultural University, Wuhan 430070, China; zsr@mail.hzau.edu.cn

**Keywords:** *Candidatus* Liberibacter asiaticus, guanosine 5′-monophosphate synthetase, virtual screening, molecular docking, enzyme activity

## Abstract

**Simple Summary:**

The spread of citrus Huanglongbing caused significant damage to the world’s citrus industry. Thermotherapy and chemical agents were used to control this disease; however, the effectiveness of these treatments is frequently inconsistent. In addition, *C*Las cannot be cultured in vitro. Therefore, structure-based virtual screening is a novel method to find compounds that work against *C*Las. This study used *C*Las GMPS as a target for high-throughput screening and selected some compounds which have a higher binding affinity to test their inhibition of *C*Las GMPS. Finally, two molecules were identified as the lead compound to control citrus HLB.

**Abstract:**

Citrus production is facing an unprecedented problem because of huanglongbing (HLB) disease. Presently, no effective HLB-easing method is available when citrus becomes infected. Guanosine 5′-monophosphate synthetase (GMPS) is a key protein in the de novo synthesis of guanine nucleotides. GMPS is used as an attractive target for developing agents that are effective against the patogen infection. In this research, homology modeling, structure-based virtual screening, and molecular docking were used to discover the new inhibitors against *C*Las GMPS. Enzyme assay showed that folic acid and AZD1152 showed high inhibition at micromole concentrations, with AZD1152 being the most potent molecule. The inhibition constant (K_i_) value of folic acid and AZD1152 was 51.98 µM and 4.05 µM, respectively. These results suggested that folic acid and AZD1152 could be considered as promising candidates for the development of *C*Las agents.

## 1. Introduction

Huanglongbing (HLB) is the most damaging disease that is threatening citrus production worldwide [1,2,3]. HLB was caused by *Candidatus* L. africanus (Laf), Ca. L. americanus (Lam), and Ca. asiaticus (Las) [4,5]. Currently, HLB has been confirmed in 51 of the 140 citrus-producing countries [6]. Current control measures against *C*Las include insecticidal control, planting *C*Las-free trees, and removing infected trees [7,8]; however, these management strategies are inadequate depending on the stage of HLB epidemiology. Despite the positive effect of using broad-spectrum antibiotics to control *C*Las [9,10,11], these antimicrobials present two significant downsides: the emergence of bacterial resistance and impact on native bacterial populations [12,13]. Small moleculars specifically targeting *C*Las have also been identified [14,15,16]. In addition, plant defense inducers or activators may help decrease the influence of HLB [17,18,19]. Nevertheless, the effectiveness of these compounds is frequently inconsistent or controversial in field conditions [20]. These problems suggest that we should focus our research on the pathogen. Considering the current shortage of new drugs, one approach to solve this problem is to apply rational drug-design techniques towards conserved metabolic pathways.

Rapid proliferation is a typical feature of bacterial infections. The de novo purine biosynthesis pathway is ultimately responsible for the generation of inosine 5′-monophosphate, and provides the adequate purine nucleotides required for DNA replication and cell division [21]. The importance of the de novo purine biosynthesis in bacterial growth has been repeatedly described. GMP synthetase (GMPS), a class-I amidotransferase belonging to the glutamine amidotransferase family, is a vital protein in the de novo purine biosynthesis, which converts xanthosine 5′-monophosphate (XMP) to guanosine monophosphate (GMP) [22]. This catalytic reaction occurs in two physically distant domains: a GATase (glutamine amidotransferase) domain that provides the required ammonia from glutamine hydrolysis and an ATPPase (ATP pyrophosphatase) domain that catalyzes the formation of the XMP-adenyl complex [23]. In *Staphylococcus aureus*, deletion of *guaA* genes resulted in guanine auxotrophy, profound abnormalities in cell morphology, and avirulence in mouse infection models [24]. In *Trypanosoma brucei*, genetic knockout of GMPS led to depletion of guanine nucleotide pools; this parasite was only rescued by additional guanine or by the expression of GMPS [25]. These studies identify GMPS as an attractive target for future drug development.

GMPS is a promising target for antibacterial drug discovery; however, only a few GMPS inhibitors have been reported [26,27,28]. Although prokaryotic GMPSs have high sequence similarities, their sensitivities to the same inhibitor are different [22,29,30]. Hence, there is a need for novel compounds targeting specific GMPSs. In this study, the structure of *C*Las GMPS was predicted using homology modeling. Drug-like molecules from the ApexBio screening library were used for virtual screening. The molecules with the highest LibDock scores were used for docking and analysis through CDOCKER. Unfortunately, CLas could not be cultured thus far. Therefore, the effectiveness of these agents against *C*Las GMPS were verified in vitro. The molecules identified in our research may serve as parent compounds for further optimization to develop agents to control the HLB disease.

## 2. Materials and Methods

### 2.1. Homology Modeling and Structure Validation

A homology model of *C*Las GMPS (UniProt ID: C6XHZ4) was built by Swiss-Model (https://swissmodel.expasy.org/interactive, accessed on 24 June 2021). The crystal structure of GMP synthase from *Neisseria gonorrhoeae* in the Protein Data Bank (PDB ID: 5TW7) was used to build the *C*Las GMPS homology model. The homology of *C*Las GMPS to *Neisseria gonorrhoeae* GMPS corresponds to 50.98% identity. This predicted model was further verified by Verify 3D [31] and Procheck [32].

### 2.2. Virtual Screening

Structure-based virtual screening was used to screen potential *C*Las GMPS inhibitors from approximately 5500 small molecule compounds available from the FDA-approved Drug Library, Natural Product Library Plus, and Inhibitor Library. The sdf file of these molecules was downloaded from ApexBio technology (https://www.apexbio.cn/screening-library.html, accessed on 24 June 2021). The LibDock section of Discovery Studio was utilized for virtual screening. LibDock is a rigid receptor-based docking suite for fast and accurate screening. LibDock employs protein site features as hotspots and then poses of rigid ligand are placed into the hotspots and matched as triplets [33]. The hotspots are used to choose the ligands to form stable complexes with the ligand-binding pocket of the receptor in LibDock module. Preparations for the use of LibDock included three steps. First, the receptor was prepared by adding hydrogen, minimizing the energy, and defining the binding site. Second, two shortcuts (“Prepare Ligands” and “Minimize Ligands”) were used to prepare the ligands. Finally, the prepared protein and prepared ligands were selected for analysis. At the end of the docking procedure, all the docked poses were ranked according to the LibDock score.

### 2.3. Molecule Docking Study

The CDOCKER module of Discovery Studio 2018 was used for molecular docking. CDOCKER is a molecular docking method which employs a CHARMm force field [34]. This force field was applied for both receptors and ligands to produce high-precision docking results. To prove the reliability of the result, two known GMPS inhibitors (DON and ACI) and the compound from the LibDock result were selected as ligands. In the docking experiment, the structure of *C*Las GMPS was used as the receptor. The best pose of each molecular binding with the receptor was estimated according to the binding energy.

### 2.4. Absorption, Distribution, Metabolism, and Excretion and Toxicity Prediction

The ADME module of Discovery Studio 2018 was employed to calculate the absorption, distribution, metabolism, and excretion (ADME) of selected compounds. The set of estimated parameters included their aqueous solubility, blood-brain barrier penetration, hepatotoxicity, human intestinal absorption, and plasma protein binding level.

### 2.5. Gene Cloning

The *guaA* gene from *Candidatus* Liberibacter asiaticus (strain psy62) was amplified by PCR using the forward primer 5′-gcgcggatccatgcacaagagagaaagatcaag-3′ and the reverse primer 5′-gcgcctcgagttattcccattcaatagttgc-3′. After treatment with the restriction enzymes *Bam* HI and *Xho* I, the guaA gene fragment was ligated to the pET28atplus expression vector and transformed into *E. coli* DH5α. Colonies were picked up and subjected to DNA sequencing. The correct plasmid was then extracted for recombinant expression.

### 2.6. Protein Expression and Purification of CLas GMPS

The plasmid pET28atplus-*C*Las GMPS was transformed into *E. coli* BL21 (DE3) and the cells were cultured in LB media supplemented with 50 µg/mL of kanamycin at 37 °C. The culture was induced by adding 0.3 mM of isopropyl-β-d-thiogalactopyranoside (IPTG) when its OD_600_ reached 0.8. The cultures were then incubated at 16 °C for 20 h.

The cells were harvested by centrifugation at 6000 rpm for 6 min at 4 °C and later suspended in buffer A (20 mM Tris-HCl, pH 8.5, 500 mM NaCl, 1 mM PMSF). Sonication of the suspension for 60 min was followed by centrifugation at 16,000 rpm for 50 min at 4 °C, which yielded a clear supernatant. His-tagged *C*Las GMPS was subsequently purified on a Ni-NTA agarose column by using an imidazole gradient. The *C*Las *G*MPS was eluted using 500 mM of imidazole and was further purified in a Superdex 200 column (GE Healthcare) equilibrated with buffer B (20 mM Tris-HCl, pH 8.5, 150 mM NaCl, 10% (*v*/*v*) glycerol) using an AKTAprime Plus System (GE Healthcare). Each eluted fraction was subjected to SDS-PAGE. Highly purified *C*Las GMPS fractions were pooled and concentrated by ultrafiltration in an Amicon-Ultra-15 Centrifugal Filter Device (Millipore, CA, USA) to 10 mg/mL. The protein concentration was quantified using the Bradford method and bovine serum albumin as a standard.

### 2.7. Enzyme Assays and Kinetics

*C*Las GMP synthase activity was continuously monitored by measuring the decrease in absorbance at 290 nm upon conversion of XMP to GMP. A VICTOR Nivo microplate reader (Perkin Elmer, MA, USA) was used to monitor the reaction rates as decrease in absorbance at 290 nm and Δε 290 was used to calculate the amount of product formed. The assay was performed in 200 μL final volume in a 96F well plate with a reaction buffer composed of 50 mM Tris-HCl, pH 8.5, 150 mM XMP, 2 mM ATP, 5 mM glutamine, 20 mM MgCl_2_, 0.1 mM EDTA, and 0.1 mM DTT. The reaction was initiated by adding 10 µg enzyme. The steady-state kinetic parameters were obtained by measuring initial velocities over a range of substrate concentrations. When the concentration of one substrate was varied, the concentrations of the other two substrates were kept at the saturating level. ATP varied over the concentration range 30 µM to 4 mM, XMP from 5 to 250 mM, Gln from 0.25 to 20 mM, and NH_4_Cl from 1 to 250 mM. The saturating concentrations of ATP, XMP, Gln, and NH_4_Cl were 2, 150, 5, and 100 mM, respectively. Each initial value represents the average of duplicate measurements, and all data were fitted to the Michaelis–Menten equation using GraphPad Prism (GraphPad Software 8.0, San Diego, CA, USA).

### 2.8. Inhibition Assay against CLas GMPS

The purchased molecules were in vitro screened. The assay was performed in a 200 µL final volume in a 96-well plate with a reaction buffer consisting of 50 mM Tris-HCl (pH 8.5), 20 mM MgCl_2_, 0.1 mM EDTA, and 0.1 mM DTT. Assays were performed at 30 °C using 10 µg *C*Las GMPS in the presence or absence of test compounds, and allowed to proceed for 60 min.

The value of K_i_ was determined at a fixed saturating concentration of ATP (1 mM) and glutamine (2 mM), different concentrations of XMP (0.04, 0.08, 0.15, 0.30, 0.40, and 0.50 mM), and in the presence of increasing concentrations of inhibitor. The concentration of folic acid was 25, 50, and 100 µM. The concentration of AZD1152 was 1, 5, and 10 µM. The concentration of DON was 1, 2, 4, and 8 µM. The concentration of ACI was between 5 and 20 µM. Each determination of K_i_ was derived from duplicate measurements and all data were analyzed using GraphPad Prism (GraphPad Software 8.0).

## 3. Results

### 3.1. Model Building and Structure Validation

Since the experimental 3D structure of the *C*Las GMPS does not exist, there was a need to predict its structure with a reasonable accuracy. Swiss–Model was employed to generate the three-dimensional structure of *C*Las GMPS using *Neisseria gonorrhoeae* GMPS (PDB ID: 5TW7) as a suitable template. The homology model of the *C*Las GMPS structure was a dimer (Figure 1).

The quality of the three-dimensional structure of *C*Las GMPS was assessed using the online platforms (Verify 3D, Procheck). The Verify 3D results showed that 94.85% of the amino acid residues had an average 3D–1D score > 0.2 (Appendix A). The Ramachandran plot was analysed by Procheck; the core region had 90.1% of the residues, 8.8% of the residues were in the allowed region, the generously allowed region had 0.7% of the residues, and 0.4% of the residues were in the disallowed region (Appendix A). These results revealed that this modeled protein had a good quality.

### 3.2. Virtual Screening

We screened 5500 molecules from the ApexBio compound database. The workflow was initiated by compound collection preparation and then subjected to LibDock. LibDock was a module of Discovery Studio 2018 and performed a high-throughput docking by aligning ligand conformations to polar and apolar receptor interaction sites. After screening, 119 compounds (LibDock score >140 and molecular weight <1000 g/mol) within the top 2% were kept after LibDock. The list of selected structures included 33 compounds from the FDA-approved-drug-library, 15 compounds from the Natural-product-library, and 71 compounds from the Inhibitor-library. All selected compounds are listed in Appendix A. Table 1 shows the top 20 ranked compounds.

### 3.3. Molecular Docking

A total of 119 structures were re-docked against *C*Las GMPS to select compounds with a relatively low binding affinity to *C*Las GMPS. Based on the binding affinity estimated by the CDOCKER docking method (-CDOCKER interaction energy >38.62 kcal/mol) in Discovery Studio 2018 (DS 2018, BIOVIA, MA, USA), the top 21 compounds were selected. CDOCKER is a grid-based molecular docking method that uses CHARMm. The receptor is held rigid, however, the ligands are allowed to flex during the process. The results were analysed to identify the ligand binding mechanisms of these molecules, and the binding affinities between receptors and ligands were calculated. Finally, we manually selected 21 hit compounds according to their binding affinity. As illustrated in Table 2, the CDOCKER potential energy of 21 moleculars were lower than the reference ligand DON (−38.6205 kcal/mol), thereby suggesting that the binding affinity of these 21 compounds with *C*Las GMPS is higher than DON.

### 3.4. ADME and Toxicity Prediction

The ADME function of Discovery Studio 2018 was used to predict the pharmacologic properties of these 21 selected compounds, including aqueous solubility level, blood–brain barrier level, hepatotoxicity, human intestinal absorption level, and plasma protein binding properties (Table 3). The solubility of each compound in water at 25 °C was predicted and it was indicated that compound529, compound1334, compound4419, and compound5481 were soluble in water. Among these four compounds, compound1334, compound4419, and compound5481 are toxic, whereas compound529 is nontoxic. These four compounds perform poorly in human intestinal absorption. Plasma protein binding properties indicated that compound529, compound1334 (AZD1152), compound4419, and compound5481 (folic acid) had weak absorption. Finally, according to low CDOCKER potential energy and good solubility, these four hit molecules were selected and purchased from the ApexBio technology with a purity of >95%.

### 3.5. Purification of CLas GMPS Protein

In this study, the GMPS gene from *Candidatus* Liberibacter asiaticus strain psy62 was cloned, and the plasmid pET28atplus-*C*Las GMPS was transformed into *E. coli* BL21 (DE3) for protein expression. *C*Las GMPS is composed of 520 amino acids and has a predicted molecular weight of 58.7 kDa. After 20 h induction in E. *coli* at 289 K, full-length His-tagged *C*Las GMPS was expressed heterologously in soluble form. *C*Las GMPS could be purified using an Ni–NTA resin affinity chromatograph following high-resolution gel filtration column (Superdex 200).

The chromatogram obtained from size exclusion chromatography (SEC) shows two peaks with the elution volumes of 47 and 70 mL in the 120 mL-Superdex 200 column (Figure 2a). According to the linear regression equation of standard protein, the size of *C*Las GMPS corresponds to a molecular mass of 119.61 kDa. This finding indicated that *C*Las GMPS forms a homodimer in solution. According to SDS-PAGE analysis, the obtained CLas protein was of high purity (Figure 2b).

### 3.6. Kinetic Characterization of CLas GMPS

For converting XMP to GMP, GMPS utilizes glutamine or ammonia as a nitrogen source. Saturation curves for *C*Las GMPS-specific activity are plotted against different concentrations of XMP (Figure 3a), ATP (Figure 3b), glutamine (Figure 3c), and NH^4+^ (Figure 3d). The kinetic properties of *C*Las GMPS are given in Table 4, according to the standard assay conditions. Fitting the sigmoidal data for the XMP saturation curve yielded values of 61.6 ± 3.9 µM for the Hill constant (K_0.5_). A similar pattern was observed for *M. tuberculosis* GMPS isozymes, which have K_0.5_ values of 45 µM [35]. The Hill slope (h) of *C*Las GMPS is 2.29, suggesting positive homotropic cooperative kinetics for XMP. Plots of initial velocity versus ATP concentration at fixed XMP and glutamine concentrations yielded: $K_{M}^{\mathrm{ATP}}$ = 258.2 ± 24.9 µM. At saturating XMP and ATP concentration, the velocity of increasing glutamine and NH4+ concentrations yielded the following values: $K_{M}^{\mathrm{Gln}}$ = 215 ± 26.76 µM, and $K_{M}^{{\mathrm{NH}}^{4+}}$ = 10.9 ± 1.9 mM. The K_M_ value for NH^4+^ is larger than that for glutamine, suggesting that glutamine may be the preferred substrate for *C*Las GMPS under the physiological conditions.

### 3.7. Inhibitory Assay against CLas GMPS Anzyme Activity

To test the inhibition effectiveness of four compounds selected by virtual screening, inhibition assay was performed. First, these four molecules were assessed for their inhibitory activities at a concentration of 1 mM against *C*Las GMPS. The inhibition mode was determined by incubation time-dependent experiments. DON and ACI were identified as irreversible inhibitors to *C*Las GMPS; DON showed 52% inhibition at 1 mM and ACI showed 15% inhibition at 1 mM. Folic acid and AZD1152 showed high inhibition against *C*Las GMPS. No exponential enzyme decay was observed against *C*Las GMPS at an extended incubation time (Figure 4). Therefore, folic acid and AZD1152 were attributed to be reversible inhibitors of *C*Las GMPS. The structure of these compounds (folic acid and AZD1152) is shown in Appendix A.

At fixed concentrations of ATP and glutamine, the inhibition constants (K_i_) were calculated to explore the mechanism of enzyme inhibition. Folic acid is a water-soluble vitamin. Recent demonstration of the strong and stable binding affinity of folic acid to the SARS-COV-2 suggests that it could be used as a potential drug for the treatment of the COVID-19 virus [37]. The K_i_ value of folic acid against *C*Las GMPS was 51.98 µM (Figure 5a). AZD1152 is a highly selective Aurora kinase inhibitor [38], The K_i_ value of AZD1152 against *C*Las GMPS is 4.05 µM (Figure 5b).

### 3.8. Analysis of Ligand Binding

The CDOCKER molecular docking module in Discovery Studio 2018 was used to identify the binding mechanism of folic acid and AZD1152. The interactions between *C*Las GMPS with folic acid and AZD1152 are illustrated in Figure 6. Schematic drawing of interactions between ligands and *C*Las GMPS is shown in Appendix A. Folic acid formed nine pairs of hydrogen bonds with VAL233, ASP234, GLN329, THR331, and ASP335 of *C*Las GMPS, and four electrostatic interactions with ASP335, GLU369, LYS376, and ARG380 of *C*Las GMPS were formed in this complex (Figure 6a,b). Four hydrogen bonds formed between AZD1152 and the residues GLY330, LEU332, GLN329, and GLY357 of *C*Las GMPS. There are three electrostatic interactions observed in the *C*Las GMPS-folic acid complex, including the π-Orbitals between folic acid and ASP234 and ASP335 of *C*Las GMPS. Moreover, one salt bridge is formed with ASP335 of *C*Las GMPS, and one halogen bond is observed with SER347 of *C*Las GMPS (Figure 6c,d).

## 4. Discussion

Citrus huanglongbing (HLB) is a destructive disease that has caused substantial crop losses. Various methods have been applied to control the spread of HLB [7,8]. However, the effectiveness of these approaches is often inconsistent under field conditions [20]. Consequently, new strategies and methods are urgently needed to prevent and control citrus HLB. Structure-based drug screening is a fast and effective method for screening specific inhibitors. GMPS is a key enzyme in the purine biosynthetic pathway. The inhibition of GMPS activity can reduce the ability of pathogen infection [24]. There is an increasing recognition that GMPS can be used as the potential antibacterial target [39]. Here, we used homology modeling to determine the structure of *C*Las GMPS and used structure-based virtual screening to identify the novel potent antimicrobial compound(s). Further inhibitory assays were performed to explore the inhibitory potential of the selected compounds.

In this study, 5500 molecules were taken from the ApexBio technology for virtual screening, followed by CDOCKER and inhibitory assay. After the screening by LibDock, 123 compounds had higher LibDock scores (LibDock score >140 and molecular weight <1000 g/mol), indicating that these 123 compounds could form a more stable complex with *C*Las GMPS. Therefore, these 123 compounds and two known GMPS inhibitors (ACI and DON) were selected for CDOCK analysis. The binding affinities of 21 compounds with *C*Las GMPS were found to be higher than that those exhibited by DON. ADME prediction was performed to predict the pharmacologic properties of these compounds, and the result indicated that four compounds (AZD1152, folic acid, Z-DEVD-FMK, and mitoxantrone-Hcl) may have better solubility.

Although bacterial enzymes have high sequence similarities, inhibition by the same inhibitor chemotype may reveal significant differences between different bacterial target enzymes [40,41]. With two known hGMPS inhibitors, Psicofuranine and decoyinine, the IC_50_ values were found to be of 17.3 and 46.5 μM, respectively [30]. Although *Pf* GMPS and hGMPS have 20−30% sequence similarity, psicofuranine at 0.5 mM showed 25% inhibition of *Pf* GMPS activity and decoynine displayed no effect on purified *Pf* GMPS [29]. Therefore, it is vital to establish a high-throughput screening method for finding novel agents that work against *C*Las GMPS. A previous study demonstrated that a single residue difference is responsible for mycophenolic acid resistance, despite a high similarity in the binding sites of the two enzymes. [42]. Accordingly, the search for inhibitory compounds toward a given target enzyme performed by virtual screening only is insufficient, and results of this search require further in vivo and in vitro experimental validation. Employing a new database that includes known compounds for screening against a novel target is very advantageous. This approach considerably reduces financial and temporal burdens in the drug discovery process. In this study, folic acid and AZD1152 were identified as inhibitors of *C*Las GMPS in the screening process. Therefore, this study is the first report that identifies folic acid and AZD1152 as *C*Las GMPS inhibitors. The K_m_ values for XMP, ATP, glutamine, and ammonium ions are shown in Table 1. *C*Las GMPS has the lowest affinity to XMP and the highest affinity towards glutamine. *C*Las GMPS was homodimer in solution. This result is consistent with the oligomeric state of GMPS proteins from *E. coli* [43], *P. falciparum* [30], and *P. horikoshii* [44]. However, GMPS proteins from humans are monomeric [34] due to a large insertion that seems to influence dimer formation [45]. The structural and mechanistic differences between GMPS enzymes from different species emphasizes the utility of the structure-based compound method for screening the potential GMPS inhibitors. Standard enzyme activity assay showed that the K_i_ values of folic acid and AZD1152 were 51.98 and 4.05 µM, respectively. Folic acid and AZD1152 showed greater potency than ACI or DON. DON showed 52% inhibition and ACI showed 15% inhibition against CLas GMPS at 1 mM. Interestingly, AZD1152 is a pro-drug of barasertib-HQPA. The CDOCKER module analysis demonstrates that the CDOCKER interaction energy of AZD1152 (−76.77 kcal/mol) was obviously lower than that of Barasertib-HQPA (−23.13 kcal/mol), which suggests AZD1152 to have a higher binding affinity with *C*Las GMPS compared with barasertib-HQPA. Identification of these two compounds certainly provides beneficial information for future structure–activity relationship studies of *C*Las GMPS inhibitors. This study provides a solid basis for using folic acid and AZD1152 as parent compounds in the future search of potent agents for controling citrus HLB.

## 5. Conclusions

This study performed a virtual screening to discover the appropriate parent compounds that inhibit *C*Las GMPS. According to the inhibitory assay against *C*Las GMPS activity, two compounds (folic acid and AZD1152) were confirmed as *C*Las GMPS inhibitors. Moreover, this study identified some drug candidates that may contribute to the design and improvement of CLas GMPS inhibitors.

## Figures and Tables

**Figure 1 biology-10-00594-f001:**
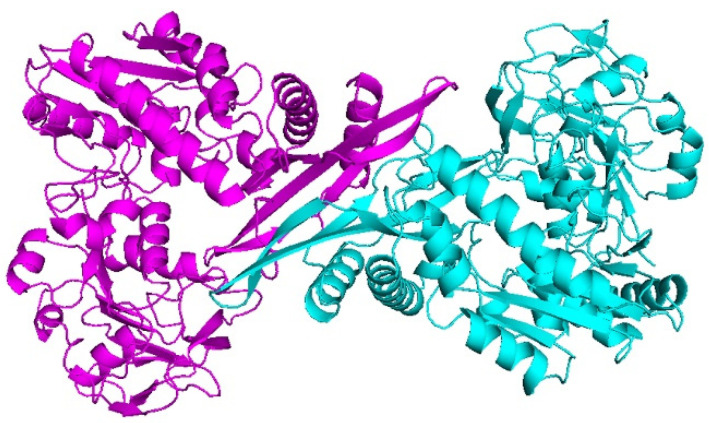
Molecular structure of *C*Las GMPS.

**Figure 2 biology-10-00594-f002:**
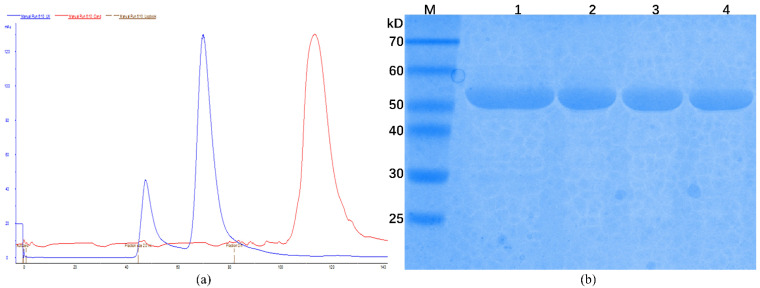
Protein purification of *C*Las GMPS. (**a**) Analysis of *C*Las GMPS with SEC; (**b**) SDS-PAGE analysis of *C*Las GMPS after purification; M: Protein marker; 1: Sample of affinity chromatography; 2–4: Sample of SEC.

**Figure 3 biology-10-00594-f003:**
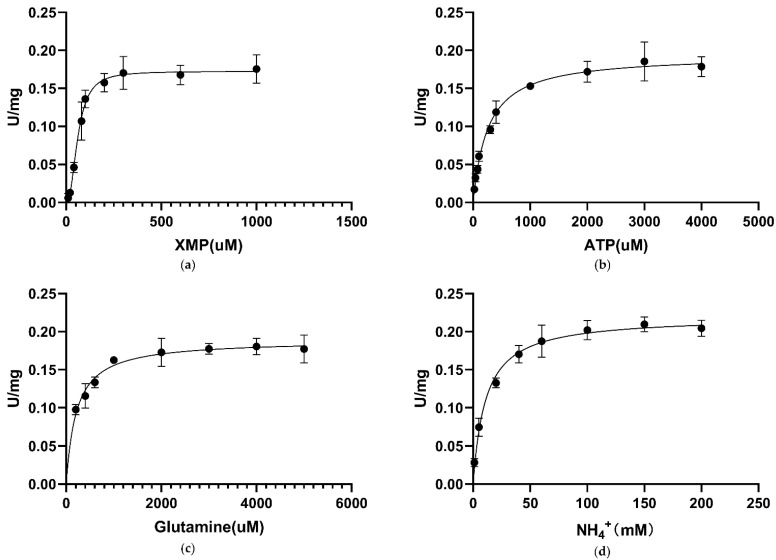
Apparent kinetic parameters for *C*Las GMPS. Specific activity as a function of the concentration. (**a**) XMP; (**b**) ATP; (**c**) glutamine; (**d**) NH^4+^.

**Figure 4 biology-10-00594-f004:**
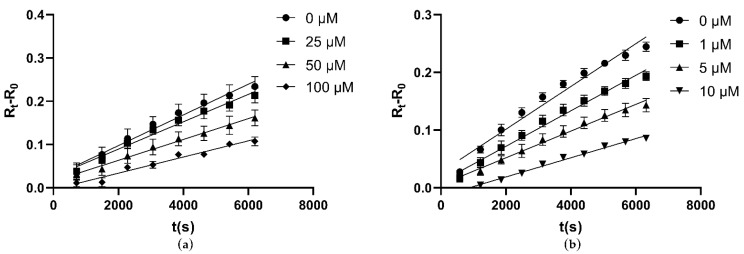
Folic acid and AZD1152 reversibly inhibit *C*Las GMPS. Elongating the reaction time, no exponential enzyme decay observed. Substrate concentrations are 1 mM ATP and 2 mM Glutamine for *C*Las GMPS. (**a**) Folic acid; (**b**) AZD1152.

**Figure 5 biology-10-00594-f005:**
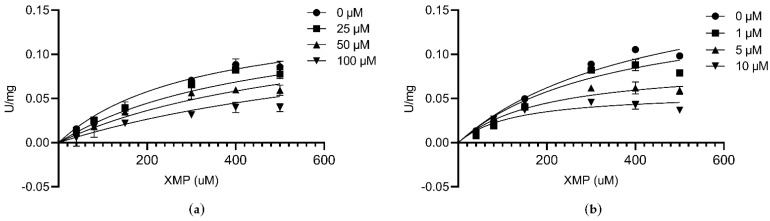
Inhibition kinetics at different concentrations of compounds by varying the XMP concentrations at a fixed ATP and glutamine concentration. (**a**) Folic acid; (**b**) AZD1152.

**Figure 6 biology-10-00594-f006:**
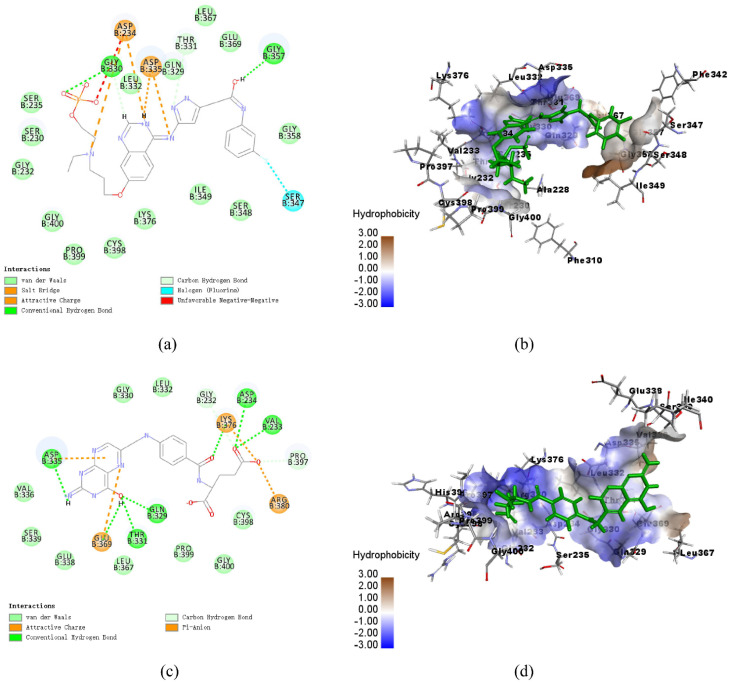
Schematic of intermolecular interaction of the predicted binding modes of *C*Las GMPS with the molecules. (**a**) 2D details of *C*Las GMPS and folic acid interaction; (**b**) 3D details of *C*Las GMPS and folic acid (blue) interaction; (**c**) 2D details of *C*Las GMPS and AZD1152 (blue) interaction; (**d**) 3D details of *C*Las GMPS and AZD1152 interaction.

**Table 1 biology-10-00594-t001:** Top 20 ranked compounds with higher LibDock.

Number	Name	CAS Number	Molecular Weight	LibDock Score
1	Compound1334	722543-31-9	587.54	191.661
2	Compound3170	1004316-88-4	776.02	183.169
3	Compound8511	81624-55-7	636.79	182.643
4	Compound3997	155213-67-5	720.9	176.273
5	Compound5520	84366-81-4	829.51	174.471
6	Compound1226	612847-09-3	551.64	174.129
7	Compound3975	852808-04-9	813.43	171.437
8	Compound2704	722544-51-6	507.56	167.324
9	Compound8477	1235034-55-5	669.79	166.047
10	Compound7726	1439399-58-2	571.57	165.201
11	Compound2503	755038-02-9	521.67	165.127
12	Compound3752	943319-70-8	532.56	164.791
13	Compound4073	641571-10-0	529.53	163.076
14	Compound1591	923288-90-8	583.99	163.076
15	Compound6234	1062159-35-6	494.59	162.396
16	Compound5313	887650-05-7	576.62	162.366
17	Compound4979	1633044-56-0	634.73	162.348
18	Compound8081	1108743-60-7	560.64	162.072
19	Compound5162	7085-55-4	742.68	161.335
20	Compound1446	356057-34-6	666.77	160.757

**Table 2 biology-10-00594-t002:** CDOCKER potential energy of compounds with *C*Las GMPS.

Number	Name	Cdocker Energy (kcal/mol)
1	Compound2265	−90.9198
2	Compound531	−89.2316
3	Compound529	−83.4206
4	Compound532	−81.3088
5	Compound1334	−76.7696
6	Compound5292	−76.5687
7	Compound4759	−75.9609
8	Compound2293	−70.4343
9	Compound4419	−67.4157
10	Compound8826	−66.4995
11	Compound1926	−65.2249
12	Compound5481	−53.8011
13	Compound2295	−51.375
14	Compound5520	−50.2805
15	Compound2476	−49.5968
16	Compound8511	−44.8984
17	Compound2240	−42.2788
18	Compound7726	−42.1659
19	Compound2101	−41.3667
20	Compound4965	−40.6106
21	Compound8477	−38.8839
22	DON	−38.6205

**Table 3 biology-10-00594-t003:** Pharmacologic properties of compounds.

Number	Name	Solubility Level	BBB Level	Hepatotoxicity	Absorption Level	PPB Level
1	Compound2265	3	4	0	3	0
2	Compound531	3	4	0	3	0
3	Compound529	4	4	0	3	0
4	Compound532	3	4	0	3	0
5	Compound1334	4	4	1	3	0
6	Compound5292	3	4	0	3	0
7	Compound4759	2	4	0	3	0
8	Compound2293	2	4	1	2	0
9	Compound4419	5	4	1	3	0
10	Compound8826	1	4	0	3	1
11	Compound1926	3	4	1	2	0
12	Compound5481	5	4	1	3	0
13	Compound2295	3	4	1	2	1
14	Compound5520	2	4	1	3	0
15	Compound2476	3	4	0	3	1
16	Compound8511	1	4	1	2	1
17	Compound2240	3	2	1	0	0
18	Compound7726	2	4	1	2	1
19	Compound2101	2	4	1	3	0
20	Compound8477	2	4	1	2	1
21	Compound4965	2	4	0	3	0

BBB, blood-brain barrier; PPB, plasma protein binding. Aqueous-solubility level: 0, extremely low; 1, very low, but possible; 2, low; 3, good; 4, optimal; 5, too soluble. BBB level: 0, very high penetrant; 1, high; 2, medium; 3, low; 4, undefined. Hepatotoxicity: 0, nontoxic; 1, toxic. Human-intestinal absorption level: 0, good; 1, moderate; 2, poor; 3, very poor. PPB: 0, absorbent weak; 1, absorbent strong.

**Table 4 biology-10-00594-t004:** Kinetic parameters of different species GMPSs.

Enzyme	XMP, K_0.5_ µM	XMP, K_m_ µM	ATP, K_m_ µM	Glutamine, K_m_ µM	(NH_4_)_2_SO_4_, K_m_ mM
*Ca. L. asiaticus*	61.6 ± 3.9	-	258.2 ± 24.89	215 ± 26.76	10.92 ± 1.30
*M. tuberculosis* ^a^	45 ± 1	-	27 ± 2	1.24 (±0.06) × 10^3^	13 ± 1
*P. falciparum* ^b^	-	16.8 ± 2	260 ± 38	472 ± 69	5.4 ± 0.8
*E. coli* ^c^	-	29	530	1000	1.0

^a^ Data are from Reference [35]. ^b^ Values are from Reference [30]. ^c^ Data are from Reference [36].

## Data Availability

All data is contained within the article. The datasets analyzed during the current study are available from the corresponding author on reasonable request.

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
