# Peer review of "Discovery of Novel GMPS Inhibitors of Candidatus Liberibacter Asiaticus by Structure Based Design and Enzyme Kinetic"

_biology, 2021, doi:10.3390/biology10070594_

Round 1
Reviewer 1 Report
Mostly minor issues are left, I will present them in the order of appearance in text.:
"Although GMPSs have high sequence similarities, the structure–activity relationships of each inhibitor are different [22,29,30]. :"
“Although GMPSs have high sequence similarities”: similarities between what species? Humans/bacteria, just bacteria, what sort of bacteria, etc.
“the structure–activity relationships of each inhibitor are different” One inhibitor has one structure-activity relationship, another one has a different structure-activity relationship, is that how this should be understood? Does it mean one can’t compare inhibitors since they belong to some sort of different QSAR equations? This is quite confusing.
Also, how does this ‘different’ QSAR of inhibitors contrasts (“although…”) with the high sequence similarity?
The sentence does not make much logical sense.
“Accordingly, novel compounds should be screened, targeting a specific GMPS”. Does it mean you take a bunch of novel compounds and screen them for actives, or you’d rather look for novel binders? Your sentence implies former. If you want to say the latter, you need to word better (e.g. “Accordingly, there is a need for novel compounds targeting specific GMPS”.)
Incidentally, how does GMPS targeting specificity is important? Is it possible that you could inadvertently develop a new antibiotic against Staph? What is the similarity between Staph/Trypanosoma and CLas GMPS? Perhaps this is connected to the similarity question in the beginning. I am not sure how to best handle this question. One thing for sure, if the binding site is very similar, the compounds you’ve discovered could also work against other, more malicious bugs! However, I also realize it’s probably out of the scope of the paper.
Verify 3D and Procheck. : references missing.
“(https://www.apexbio.cn/screening-library.html). “ the link doesn’t work
“LibDock is rigid…” -> LibDock is a rigid…”
“for fast and accurate screening, LibDock uses” _> ““for fast and accurate screening. LibDock uses”
“First, the receptor was prepared, which involves” -> “First, the receptor was prepared, which involved”; better yet: ““First, the receptor was prepared by”
Why did you describe in detail preparation of the protein/ligand fro LibDock, and do not do the same for CDOCKER? Should be consistent. IMHO, if this is a standard protocol using tools in the program itself, theses steps could be left out altogether. I think the preparation of receptor/ligand becomes more relevant when outside tools are used. You decide. But please be consistent.
Molecular dynamics simulations: was the actual engine to produce MD was called CHARMM?
Incidentally, sometimes you use Discovery Studio 2019, other times 2018. That is not an error?
“CDOCKER … employs CHARMm” CHARMm force field, or something else?
“according to binding affinity” -> “according to the binding affinity”
“wheres compound529 is nontoxic.” there is no word “wheres” ! did you mean
“while”? “whereas”?
“The RMSD plot showed that most of the protein-ligand complexes were found to be stable up to 100 ps during the molecular dynamics simulation as shown in Figure 7.”
This implies that the protein becomes unstable after 100 ps, and that would mean the model fails! However, I do think that 2 Angs overall RMSD is reasonable, especially for a homology model of a protein with quite a few unstructured elements/loops according to Figure 1. So please modify your text accordingly.
Incidentally, Figure 1 is repeated twice in the manuscript.
What’s “conf1” in Fig. 7?
Reviewer 2 Report
MAJOR ISSUE:
The “Molecular Dynamics simulations” performed by the authors are only 300 ps, which is TOO short (less than even one nanosecond). When performing MD simulations of proteins or protein models (with or without substrates/ligands bound), the simulation time needs to be ample enough for the systems to reach convergence; this means the environment itself (solvent fluidity, temperature and pressure equilibrium, i.e. the process of equilibration), as well as the protein (i.e. no slow, large-scale movements are expected to occur and the protein system has reached its final state). This time scale depends on the system’s properties (and convergence can be checked with various methods such as RMSD-based clustering or essential dynamics), but, at minimum, requires at least 25 nanoseconds (25000 ps) of simulation for small-to-medium proteins (BPTI, crambin, etc). For larger proteins such as GMPS, this time scale can increase up to 100 ns, or even exceed that.
If the authors want to keep MD simulations in the manuscript, they should perform a simulation that meets the following criteria: ample equilibration time (monitored by evaluating the temperature, pressure and volume curves), and either one large production simulation, for which protein convergence can be evaluated with RMSD or Essential Dynamics, or alternatively, multiple (at least three) short (~25 ns) simulations of the system, starting at randomized configurations, that can display convergence among them (i.e. the system does not deviate much among different simulations). The above are the standard parameters required for a valid MD report and are considered prerequisites in publications dealing with protein simulations.
If, however, the authors have no access to the computational resources required to perform such simulations, they should skip this step entirely and remove the MD simulations from their methods and results, especially since they already have experimental results validating, to some extent, their theoretical model.
MINOR ISSUES:
Point 1: There still exist a few syntax errors that need to be corrected, for example:
- Abstract, page 1, row 11: “…no effective HLB-easing approach is available exists once the trees become infected…”. Keep “is available” or “exists”, but not both!
- Abstract, page 1, row 12: add an “is” verb after (GMPS), i.e. “Guanosine 5′-monophosphate synthetase (GMPS) is a key enzyme …”
These, as well as any other syntax errors found in the text, should be corrected.
Point 2: In the Methods subsection 2.2, “Virtual Screening”, page 2, row 90, the authors state that “the receptor was prepared, which involves removing water and other heteroatoms…”. While this is true for actual crystal structures, it does not apply to homology models which, typically, have only protein atom coordinates, not heteroatoms. This sentence should be removed.
Round 2
Reviewer 2 Report
The authors have addressed my concerns, and through the changes they have made, they have improved the overall quality of the manuscript. This paper is, in my opinion, in a state acceptable for publication.
Author Response
Thanks for your suggestion.